# AUTOMATED FEATURE LABELING WITH TOKEN-SPACE GRADIENT DESCENT

**Julian Schulz**
Independent
mail@julianschulz.eu

**Seamus Fallows**
Independent
seamusfallows1@gmail.com

## ABSTRACT

We present a novel approach to feature labeling using gradient descent in token-space. While existing methods typically use language models to generate hypotheses about feature meanings, our method directly optimizes label representations by using a language model as a discriminator to predict feature activations. We formulate this as a multi-objective optimization problem in token-space, balancing prediction accuracy, entropy minimization, and linguistic naturalness. Our proof-of-concept experiments demonstrate successful convergence to interpretable single-token labels across diverse domains, including features for detecting animals, mammals, Chinese text, and numbers. Although our current implementation is constrained to single-token labels and relatively simple features, the results suggest that token-space gradient descent could become a valuable addition to the interpretability researcher's toolkit.

## 1 INTRODUCTION

Recent work in mechanistic interpretability has made significant progress in decomposing neural networks into interpretable features. Methods like Sparse Autoencoders (SAEs) have shown success in identifying meaningful patterns of activation in language models – from basic syntactic features to high-level semantic concepts (Bricken et al., 2023; Braun et al., 2024; Huben et al., 2024; Rajamanoharan et al., 2024). These methods extract features that represent specific activation patterns, with each feature having an activation value for every token in a text.

Standard approaches to labeling these features involve either manually identifying specific tokens or contexts that trigger high activation levels or prompting language models with examples of activating contexts and asking them to generate hypotheses about what the feature represents (Bills et al., 2023; Paulo et al., 2024). In Templeton et al. (2024), they also considered the reverse problem of searching for specific features in an SAE by using targeted prompts relating to the concept of interest and inspecting the features that activate most strongly for specific tokens in those prompts.

In this work, we explore an alternative approach to feature labeling. Rather than using language models to generate hypotheses about feature meanings, we reformulate the task as an optimization problem in the space of tokens. By having a language model evaluate candidate descriptions against observed feature activations, we create a differentiable pipeline for searching through possible labels. Our experiments test this approach on several different features, examining both its capabilities and limitations.

The key insight behind our approach is that predicting whether a given token matches a feature description is a much simpler task for language models than generating accurate hypotheses about feature meanings. Using the language model only for this simpler classification task, we can leverage gradient descent to efficiently search the space of possible descriptions.

## 2 METHOD

One way to think about feature labels is as descriptions that would enable a human to accurately predict where a feature activates. Given a text and a feature label such as "animal", a human can

identify which tokens match this concept, and we can evaluate the quality of this label by comparing these predictions to the actual activation pattern of the feature (Figure 1).

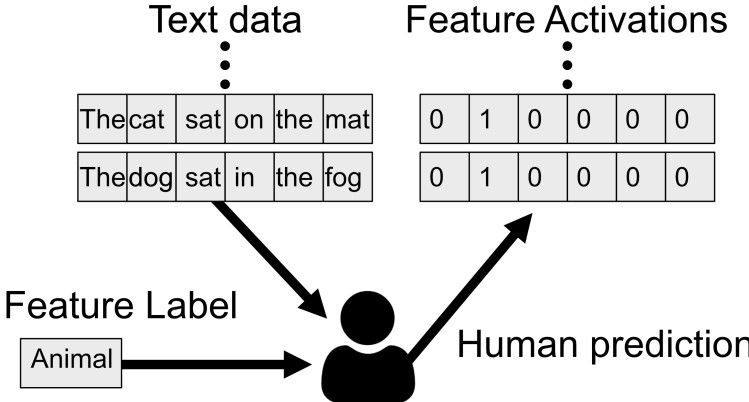

Figure 1: A feature label can be evaluated by how well it enables prediction of feature activations. Here, a human uses the label "Animal" to predict which tokens in the input text will have high feature activation values.

This framing suggests a natural metric for feature labels: how accurately do they enable prediction of feature activations. By replacing the human with a language model, we can make this evaluation process differentiable (Figure 2). This allows us to perform gradient descent on the feature label itself, directly optimizing it to enable accurate predictions. This is reminiscent of feature visualization techniques (Olah et al., 2017) in computer vision, where input signals are optimized to maximize neuron activation. Here, we optimize a probability distribution over tokens so that the resulting label best predicts feature activations.

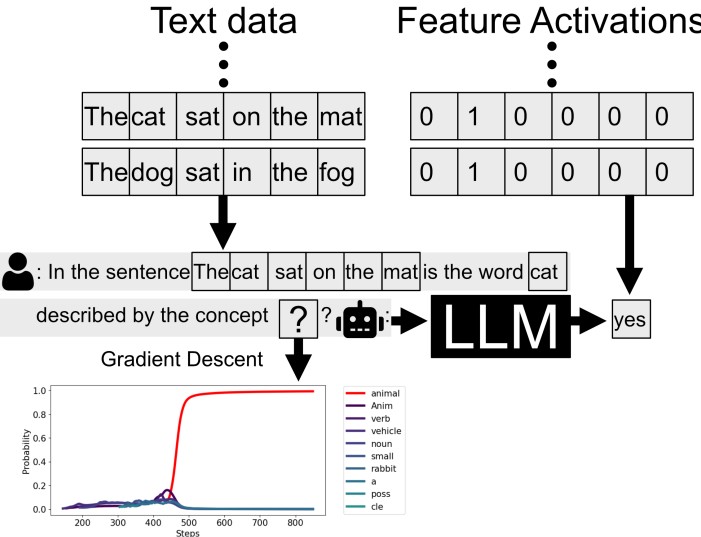

Figure 2: Our method replaces the human evaluator with a language model, making the process differentiable. The bottom plot shows the optimization trajectory in token-space, where the probability mass concentrates on the token "animal" after several hundred steps.

## 2.1 SETUP

Our method assumes that we have a feature that assigns a numerical activation value to each token in a corpus of text. To evaluate potential feature labels, we use a language model as a binary classifier,

determining whether each token matches a given feature description. The classification process uses a structured prompt with three components:

- A system prompt that defines the task as binary classification
- The text context, target token, and candidate feature label
- A restricted output format ("Rating: 0" or "Rating: 1")

Specifically, we use the following prompt structure:

```
[System] Your task is to assess if a given word from some text represents
    a specified concept in the context of the text. This could be a
    concept related to the meaning of the word, or its structural usage
    in the text. Provide a rating based on this assessment: If the word
    represents the concept, respond with 'Rating: 1'. If the word does
    not represent the concept, respond with 'Rating: 0'. Think carefully
    about if the concept applies to the word in the context of the text.
    Be confident.

[User] The text is: "{sentence}". From this text, is the word "{token}"
    an example of the concept "{candidate label}"?

[Assistant] Rating:
```

We extract the model's prediction by comparing the logits of the tokens "0" and "1" immediately following this prompt. During training, we iterate through the corpus, evaluating each token in each sentence against our current candidate label. To ensure balanced training data, we up-weight tokens where the feature is active, so that the training batches contain an equal number of active and inactive feature examples. For simplicity, we used synthetic feature activation data with the activation on each token being either $1$ (active) or $0$ (inactive)[1]. We used Meta-Llama-3-8B-Instruct as the evaluator (AI@Meta, 2024).

## 2.2 LABEL REPRESENTATION

We represent candidate labels using a vector of label logits $\mathbf{v}$ in token-space (dimension $d_{\text{vocab}}$) rather than in embedding space (dimension $d_{\text{model}}$). This vector is the parameter that we optimize through gradient descent. We apply softmax to obtain a probability distribution $\mathbf{p}$ over tokens,

$$\mathbf{p} = \text{softmax}(\mathbf{v}). \tag{1}$$

This probability distribution is then converted to an embedding by multiplying with the transformer's embedding matrix $E$,

$$\mathbf{e} = E\mathbf{p}. \tag{2}$$

This differs from standard transformer operation, where typically only one-hot vectors are used, effectively selecting a single row of the embedding matrix. Our approach allows for a "superposition" of tokens during optimization, where multiple tokens can contribute to the label's meaning. The convergence plot in Figure 2 shows a subset of these token probabilities $\mathbf{p}$ over the course of optimization.

Our approach to optimizing the input to an LLM is similar to that of Rumbelow & Watkins (2023). However, a key difference is that they optimized over a vector in the *embedding space*, incorporating a regularization term to minimize the distance to the nearest valid token embedding, along with projection onto the nearest valid token embedding during optimization, to ensure the optimized inputs remained close to legal tokens. Our technique uses a single regularization term corresponding to the entropy of the probability distribution $\mathbf{p}$, as we explain below. See also Ebrahimi et al. (2017) for work in a similar vein.

---

[1]Real SAE activations take values in $\mathbb{R}$ rather than being restricted to $\{0, 1\}$. However, they can be discretized using a threshold. Alternatively, one could modify the prompt to have the evaluator model predict the activation value directly.

## 2.3 LOSS FUNCTION

Our loss function combines three terms:

$$L(\mathbf{v}) = L_{\text{acc}}(\mathbf{v}) + \lambda_{\text{ent}} L_{\text{ent}}(\mathbf{v}) + \lambda_{\text{kl}} L_{\text{kl}}(\mathbf{v}) \tag{3}$$

where $\lambda_{\text{ent}}$ and $\lambda_{\text{kl}}$ are hyperparameters.

The accuracy loss measures how well the label predicts feature activations,

$$L_{\text{acc}}(\mathbf{v}) = -\frac{1}{n} \sum_{t=1}^{n} [f(t) \log m(t, \mathbf{p}) + (1 - f(t)) \log(1 - m(t, \mathbf{p}))], \tag{4}$$

where $f(t)$ is the feature activation for token $t$ and $m(t, \mathbf{p})$ is the LLM's prediction given our current label distribution $\mathbf{p}$. This term ensures that our label correctly identifies which tokens have high feature activations.

The entropy loss is computed on the label probability distribution,

$$L_{\text{ent}}(\mathbf{v}) = -\sum_{i} p_i \log p_i. \tag{5}$$

This term encourages concentration on a single token, pushing against solutions that distribute probability mass across many tokens. Although such distributed representations might achieve good prediction accuracy, they would not be human-interpretable.

The KL-divergence loss compares our label distribution to the LLM's expected token distribution $\mathbf{q}$ at the label position,

$$L_{\text{kl}}(\mathbf{v}) = \sum_{i} p_i \log(p_i/q_i). \tag{6}$$

This term encourages linguistic naturalness by penalizing tokens that would be unexpected as feature labels in the prompt context.

## 3 RESULTS

### 3.1 SUCCESSFUL CASES

Our method successfully identified accurate single-token labels across several test cases. Figure 3 illustrates the optimization trajectories of label token probabilities for four distinct synthetic features.

**Animals.** In the animal detection case (Figure 3A), the dataset consisted of natural language sentences containing animal references, the feature being active on tokens representing animals. The optimization process converged decisively to the token "animal" after approximately 500 steps. Notably, other tokens that temporarily gained significant probability during training were semantically related, such as "Anim" and "rabbit", suggesting that the optimization process explored the relevant semantic space.

**Mammals.** For mammal detection (Figure 3B), the dataset consisted of a balanced set of mammals and non-mammal animals with the feature active only on mammals. The optimization converged to the token "mamm" after roughly 300 steps. Interestingly, despite the existence of a "mammals" token in the LLama 3 tokenizer, this complete form never gained significant probability mass during training. We discuss the case of an unbalanced dataset in the next section.

**Chinese text.** The Chinese text detection case (Figure 3C) presented a particularly interesting result. Trained on a mixture of English and Chinese text, the optimization initially appeared to stagnate in a superposition of less informative tokens before suddenly converging to "中文" (Chinese characters meaning "Chinese") at around step 280. It is interesting to note that the optimization converged to Chinese characters despite the entire prompt being formulated in English. The English token "Chinese" briefly gained prominence around step 40 but subsequently lost probability mass to other tokens before the final convergence.

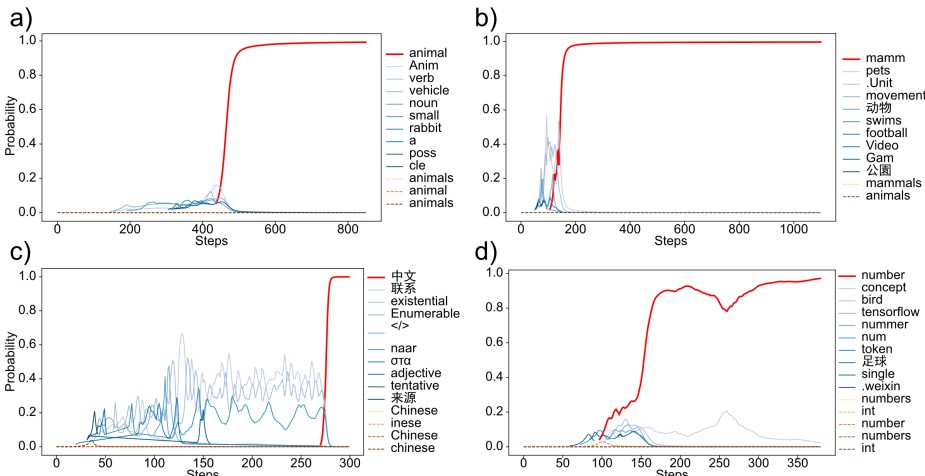

Figure 3: Optimization trajectories showing token probability distributions over training steps for successfully labeled features. Each panel represents a different synthetic feature trained on distinct text corpora. The winning token is shown in red, while the next 9 highest-probability tokens during optimization are shown in blue shades. Alternative valid descriptive tokens are traced with red dotted lines. (a) Animal detection feature: converges to "animal" after 500 steps, with semantically related tokens (e.g., "Anim", "rabbit") showing temporary prominence. (b) Mammal-specific feature: converges to "mamm" when trained on a balanced dataset of mammal and non-mammal animals. (c) Chinese text detection: shows sudden convergence to "中文" (meaning "Chinese") at step 280. (d) Number detection: quick convergence to "number" for a feature active on both numerical digits and number words, with related tokens like "num" and "nummer" appearing during optimization. Training data and hyperparameters are detailed in appendix A.1.

**Numbers.** In the number detection case (Figure 3D), the feature was trained to activate on both numerical digits and number words (e.g., "9" and "thirteen"). The optimization quickly converged to the token "number", while semantically related tokens like "num" and the German word "nummer" (meaning "number") showed temporary prominence during training.

## 3.2 FAILURE CASES AND LIMITATIONS

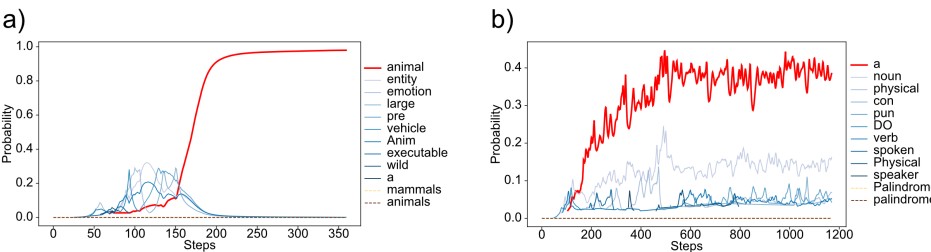

Figure 4: Optimization trajectories for unsuccessful feature labeling attempts. Format matches Figure 3, with winning tokens in red and top competing tokens in blue. (a) Failed mammal detection: when trained on natural text without balanced sampling, the optimization converges to the broader category "animal" instead of the intended mammal-specific label. (b) Failed palindrome detection: optimization defaults to the simple token "a", due to base model's inability to reliably classify palindromes even with correct labeling. Complete training data and hyperparameters are provided in appendix A.1.

While these results demonstrate the method's ability to identify meaningful single-token descriptions across various feature types, some limitations should be noted. The resulting labels sometimes

require additional interpretation – for example, "mamm" is not immediately obvious as representing "mammal" without context. Similarly, the Chinese character label requires translation for non-Chinese speakers.

Two features that we tested failed to result in the correct labels. These were a "palindrome" feature and our initial attempt at finding a "mammal" label when training on unbalanced data. These are shown in Figure 4.

When training on text containing both mammals and non-mammal animals (Figure 4A), the method consistently converged to the broader category "animal" rather than the intended "mammal" distinction. This failure was traced to our sampling strategy – due to upsampling of tokens where the feature was active, mammal tokens were frequently sampled, while non-mammal animal tokens remained rare in the training data. This allowed the optimization to achieve low accuracy loss while misclassifying non-mammal animal tokens. We verified this explanation by creating a modified dataset with a substantial proportion of non-mammal animal tokens, which successfully yielded the "mamm" label shown in Figure 3B.

We found that the failure to arrive at the correct "palindrome" label (Figure 4B) was simply due to the LLM's inability to correctly classify palindromes even when provided directly with the correct label.

Additionally, these results were obtained with some hyperparameter tuning for each case (complete hyperparameter settings are provided in appendix A.1). However, we did not perform a systematic search for a single set of hyperparameters that work across all cases, so it is unclear whether case-by-case tuning is necessary.

These failure cases provide insights into the method's limitations and potential paths for improvement, which we discuss in the following section.

## 4    DISCUSSION

### 4.1    LIMITATIONS AND POSSIBLE IMPROVEMENTS

**Single-token labels**    Our experiments revealed several limitations of the current implementation that could be addressed through future improvements. Most notably, many feature descriptions cannot be adequately captured by single tokens. For example, in our mammals example, the method converged to "mamm", which requires interpretation as the beginning of "mammal". This limitation could potentially be overcome by generalizing the method to optimize multiple token positions simultaneously.

**Hierarchical categories.**    We also observed challenges with hierarchical categories, as shown in the mammal detection failure case. When insufficient examples of the distinguishing cases are present (like non-mammal animals), the method tends to default to broader categories (like "animal"). This could be addressed through more sophisticated sampling strategies. We could sample equally from four distributions: (1) tokens where the feature is active but misclassified as inactive, (2) tokens where the feature is inactive but misclassified as active, (3) tokens correctly classified as active, and (4) tokens correctly classified as inactive. This balanced sampling approach would ensure that misclassifications receive appropriate weight in the optimization process.

**Model capability.**    Another limitation comes from the capabilities of the evaluation model. As demonstrated in the palindrome example, if the model cannot reliably classify instances of a feature even when given the correct label, our method cannot converge to an accurate description. However, this is a limitation for all current methods that rely on a language model's ability to recognize relevant concepts.

**Hyperparameter tuning.**    Currently, our method still requires human intervention to tune hyperparameters for different features. If no universal hyperparameter set is found, this could still be automated by implementing systematic hyperparameter sweeps that continue until finding parameters that achieve both convergence to a single token and low accuracy loss.

**Prompt format.**    Our current prompt format is limited to features that are clearly active or inactive on individual tokens or contexts. Some features, such as those that detect repetition patterns or long-range dependencies, would require modifications to the prompt structure to capture their activation patterns accurately.

**Computational cost.**    Finally, the computational intensity of running a separate optimization process for each feature makes it relatively expensive compared to direct LLM-based labeling approaches. However, reasoning models might have to generate long chain-of-thought sequences in order to identify some complex features, which could be comparable or even more computationally expensive than a gradient descent-based approach.

## 4.2    APPLICATIONS

This paper serves as a proof-of-concept demonstration for token-space gradient descent in automated feature labeling. If developed further with the improvements outlined above, it could become a valuable tool for labeling features found by various interpretability methods, such as Sparse Autoencoders, Transcoders, and Crosscoders.

The practical utility of this method will ultimately depend on whether it can produce better labels more efficiently than direct LLM-based approaches. While there are reasons to be skeptical given the rapid improvements in LLM reasoning capabilities and the computational cost of our training process, the method may still have important applications in the context of AI safety research.

Specifically, when using interpretability for safety analysis – such as identifying potentially dangerous behavioral patterns in models – relying solely on LLMs to generate feature labels creates a potential vulnerability. A misaligned LLM might intentionally provide poor labels for safety-relevant features, effectively *sandbagging* the interpretability process. By removing the hypothesis generation role from the LLM and instead using gradient descent, our method could potentially provide a more robust approach to safety-critical interpretability work.

## ACKNOWLEDGMENTS

We thank Dmitrii Krasheninnikov for helpful discussions and comments on earlier drafts of this paper. This research was supported by a Long-Term Future Fund grant to Seamus Fallows and by the Open Philanthropy Career Development and Transition Funding program for Julian Schulz.

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

# A    APPENDIX

## A.1    HYPERPARAMETERS AND TRAINING DATA FOR SYNTHETIC FEATURES

Below we provide samples from each dataset used for the experiments. In each dataset, the tokens where the feature under consideration is active are printed in bold. The hyperparameters used for every shown result are listed in table 1.

> The **cat** chased the **mouse**.
> A **dog** barked at the mailman.
> The **bird** sang a sweet song.
> A **frog** leaped into the pond.
> The **duck** swam in the lake.
> A **bear** hibernated in the cave.

Figure 5: Animal Text Dataset: Natural language sentences containing animals. The feature is active (bold) on words referring to animals.

**fox**
**wolf**
crow
crab
cherry
Accordion

Figure 6: Mammal Words Dataset: Single-word examples including mammals and non-mammals. The feature is active (bold) only on mammal words.

The gauge group of the standard model is SU(3)XSU(2)XU(1).
他一到，我们就开始
The dog ran after the man.
我在想你今天会不会来
Hello darkness my old friend
现在大地变得无形空虚，黑暗笼罩着深渊的表面。

Figure 7: Chinese/English Mixed Text Dataset: Alternating sentences in English and Chinese. The feature is active on all Chinese characters.

apple
book
**3**
**4**
**five**
**six**

Figure 8: Number Words Dataset: Mix of digits and words. The feature is active (bold) on both numerical digits and written number words.

The **cat** chased the **mouse**.
A **dog** barked at the mailman.
The bird sang a sweet song.
A frog leaped into the pond.
The duck swam in the lake.
A **bear** hibernated in the cave.

Figure 9: Mammal Text Dataset: Natural language sentences containing both mammals and non-mammal animals. The feature is active (bold) only on mammal references.

The **civic** center is hosting a charity event this weekend.
I love to **kayak** on the lake during the summer months.
The carpenter made sure the **level** before installing it.
Please **level** the playing field by giving everyone equal opportunities.
The **radar** detected a large storm system approaching from the west.
Please **refer** to the user manual for detailed instructions.

Figure 10: Palindrome Text Dataset: Sentences containing palindrome words. The feature is active (bold) on words that read the same forwards and backwards.

Table 1: Experiment configurations and hyperparameters

| Plot | Dataset | Batch Size | Epochs | LR | Loss Coefficients | | |
|------|---------|------------|--------|-----|----------|---------------|---------|
| | | | | | accuracy | kl-divergence | entropy |
| Fig. 3 a) | Animal Text (Fig. 5) | 10 | 50 | 0.03 | 1 | 0.2 | 0.2 |
| Fig. 3 b) | Mammal Words (Fig. 6) | 10 | 100 | 0.2 | 1 | 0.1 | 0.3 |
| Fig. 3 c) | Chinese & English text (Fig. 7) | 25 | 100 | 0.5 | 1 | 0.05 | 0.25 |
| Fig. 3 d) | Number Words (Fig. 8) | 20 | 10 | 0.1 | 1 | 0.2 | 0.2 |
| Fig. 4 a) | Mammal text (Fig. 9) | 20 | 10 | 0.1 | 1 | 0.2 | 0.2 |
| Fig. 4 b) | Palindrome Words (Fig. 10) | 15 | 15 | 0.1 | 1 | 0.2 | 0.2 |

Table 2: Detailed experimental configurations for each feature labeling experiment shown in Figures 3 and 4. The Dataset column describes the type of text used for training. Batch Size indicates the number of tokens evaluated in each training step. Epochs shows the total number of passes through the training data. LR represents the learning rate used for gradient descent. The Loss Coefficients columns show the relative weights assigned to each component of the loss function: prediction accuracy, KL-divergence (for linguistic naturalness), and entropy (for encouraging single-token convergence). All experiments used Meta-Llama-3-8B-Instruct as the evaluator model.

