# OpenReview forum: "Automated Feature Labeling with Token-Space Gradient Descent"
_ICLR.cc/2025/Workshop/BuildingTrust — BuildingTrust_

### Official Review · Reviewer_6LmN · 2025-02-23
**Well-Scoped Proof of Concept**

**Rating:** 7
**Confidence:** 4

**Review:**

This work is well-scoped and has a clear understanding of the strengths and weaknesses of its method. I think that it is valuable to prove that differentiable methods can effectively succeed at feature labeling, even though there are obvious computational problems that make scaling this method incredibly difficult. The paper is well-written, and the label representation choices are smart and effective. I have a couple of thoughts for potential ways to improve the clarity of the results:
- It would be useful to do a more direct contrast between the paper results and the effectiveness of prompting LLMs to provide feature explanations. For example, for the datasets you use, what is the distribution of LLM naive responses and how would it categorize features? It is worthwhile to have a baseline to show how much more effective your feature classification method is, especially in terms of confidence of classification. This feels like a fairly straightforward experiment to run.
- I am interested in seeing how the hyperparameter for the entropy loss informs the outcomes of your model predictions. I could imagine that having a large weighting factor toward collapsing the probability distribution over token space could incentivize the model to develop a high probability for an answer, even when that answer is incorrect. What happens in the failed cases when this parameter is smaller?
- I think the language around "candidate descriptions" is slightly confusing. This phrasing, alongside the labels being fixed in the legends of all training graphs, makes it seem like you are optimizing over a fixed number of potential labels (that might be hand-picked), instead of all potential labels in the token space. I think that this could be made less ambiguous.
- A small note: there seems to be a typo in Figure 1.

---

### Official Review · Reviewer_Wgvm · 2025-03-03
**An interesting idea, but results are primarily a couple of hand-picked examples**

**Rating:** 6
**Confidence:** 3

**Review:**

### Summary:
The paper presents an approach to feature labeling which uses gradient descent on tokens to find the token that most closely matches the given feature.

### Strengths:

- I am not familiar with the literature, but it seems like an interesting idea.
- I found the paper difficult to understand the first time reading through. It would benefit from a clear figure of the training loop and which parts are being optimized and iterated over, etc. Pseudocode could also be helpful.

### Weaknesses:

- Ebrahimi et al. (2017) is mentioned as similar in the related work. How does the work relate to/differ from this?
- Is the feature that is being labelled a feature in an actual model? If so, what model? Or is it essentially being simulated in a rule-based way? This should be made clearer.
- It would be helpful to include an ablation of the different loss terms.
- Would benefit greatly from a meaningful metric of the performance rather than a couple of handpicked examples.

---

### Decision · Program_Chairs · 2025-03-04

Accept